# Robotic Surgery and Functional Esophageal Disorders: A Systematic Review and Meta-Analysis

**DOI:** 10.3390/jpm13020231

**Published:** 2023-01-27

**Authors:** Sara Vertaldi, Anna D’Amore, Michele Manigrasso, Pietro Anoldo, Alessia Chini, Francesco Maione, Marcella Pesce, Giovanni Sarnelli, Giovanni Domenico De Palma, Marco Milone

**Affiliations:** 1Department of Clinical Medicine and Surgery, University of Naples “Federico II”, 80131 Naples, Italy; 2Department of Advanced Biomedical Sciences, University of Naples “Federico II”, 80131 Naples, Italy

**Keywords:** robotic fundoplication, laparoscopic fundoplication, reflux, hiatal hernia, functional outcomes

## Abstract

The functional disease of the esophago-gastric junction (EGJ) is one of the most common health problems. It often happens that patients suffering from GERD need surgical management. The laparoscopic fundoplication has been considered the gold standard surgical treatment for functional diseases of the EGJ. The aim of our meta-analysis is to investigate functional outcomes after robotic fundoplication compared with conventional laparoscopic fundoplication. A prospective search of online databases was performed by two independent reviewers using the search string “robotic and laparoscopic fundoplication”, including all the articles from 1996 to December 2021. The risk of bias within each study was assessed with the Cochrane ROBINS-I and RoB 2.0 tools. Statistical analysis was performed using Review Manager version 5.4. In addition, sixteen studies were included in the final analysis, involving only four RCTs. The primary endpoints were functional outcomes after laparoscopic (LF) and robotic fundoplication (RF). No significant differences between the two groups were found in 30-day readmission rates (*p* = 0.73), persistence of symptomatology at follow-up (*p* = 0.60), recurrence (*p* = 0.36), and reoperation (*p* = 0.81). The laparoscopic fundoplication represents the gold standard treatment for the functional disease of the EGJ. According to our results, the robotic approach seems to be safe and feasible as well. Further randomized controlled studies are required to better evaluate the advantages of robotic fundoplication.

## 1. Introduction

The functional disease of the esophago-gastric junction (EGJ) is one of the most common health problems, affecting more than 50% of the world’s population and resulting in a serious deterioration of quality of life with important economic implications [1]. Medical treatment with a proton pump inhibitor (PPI) can help control reflux symptoms, but on the other hand, it implies cost-effective, long-lasting medicine-based treatment. It often happens that GERD patients do not achieve complete control of the symptoms, needing surgical management [2].

The laparoscopic fundoplication has been performed with patient satisfaction since its introduction during the twentieth century, becoming the gold standard for the surgical treatment of functional disease of the EGJ [3,4]. 

Additionally, after the first robotic-assisted Nissen fundoplication (RALF) reported by Cadiere in 1999 [5], it has been debated if the robotic approach could improve surgical outcomes due to the three-dimensional view and the enhanced manipulation of instruments [6] compared with the conventional laparoscopic fundoplication (CLF). Several previous studies have demonstrated the safety and feasibility of a robot-assisted approach in this setting [7,8,9,10]. 

The aim of our meta-analysis is to investigate the functional outcomes after minimally invasive surgery, both laparoscopic and robotic, for the treatment of functional disease of the EGJ.

## 2. Materials and Methods

### 2.1. Literature Search and Eligibility Criteria

This systematic review complied with PRISMA (preferred reporting items for systematic reviews and meta-analyses) reporting standards [11] and was developed in line with MOOSE (meta-analysis of observational studies in epidemiology) guidelines [12].

A prospective search of Embase, PubMed, SCOPUS, and Web of Science was performed using the search string “robotic and laparoscopic fundoplication”. 

The analysis included all the articles from 1996 to December 2021. The last search was performed in January 2022. 

Case reports, case series without a control group, indexed abstracts of posters and podium presentations at international meetings, and non-English articles were excluded. Systematic reviews and meta-analyses were only consulted to identify additional studies of interest. In addition, the reference lists of the retrieved studies were manually reviewed. In cases of overlapping series in different studies, only the most recent article was included. Publications with no data about functional results after minimally invasive fundoplication were also excluded.

The research question was structured using the PICO (problem/population, intervention, comparison, and outcome) framework. The populations of interest included patients affected by functional esophageal disease (GERD, hiatal hernia, or paraesophageal hernia). The intervention was robotic fundoplication, and the comparator was laparoscopic fundoplication. The functional outcomes after surgery were analyzed: 30-day readmission; persistent symptomatology at follow-up, including delayed gastric emptying; postoperative pyrosis or dysphagia; disease recurrence; need for reintervention.

The literature search and study selection were performed independently by two reviewers (S.V. and A.D.), showing a high level of inter-reader agreement (κ = 1). In case of disagreement, a third investigator (Mi.Ma.) was consulted, and an agreement was reached by consensus. 

### 2.2. Data Extraction and Assessment of Risk of Bias in Included Studies

The titles and abstracts were screened and reviewed independently by S.V. and A.D., followed by full-text reading. In addition, ineligible studies were excluded after full-text reading. The data extraction was conducted independently and in duplicate by the two reviewers. Further, the data extraction form was created in accordance with the guidelines in the Cochrane Handbook for systematic reviews of interventions by the consensus of both reviewers. 

The following data were extracted from each included study: first author, year of publication, study design, period of study, surgical indication, sample size, number of patients in each surgical group, gender, mean age, mean BMI, type of intervention (Nissen, Dor, Toupet, or no fundoplication), redo surgery, operative time, 30 days readmission, mean follow-up, persistence of symptomatology at follow-up, complaining of delayed gastric emptying, pyrosis, or dysphagia, needing of reintervention. The data extracted from studies were then separated into the following sections: study characteristics, population characteristics, intervention characteristics, and functional outcomes.

Additionally, after data extraction was completed, the risk of bias within each study was assessed. 

The Cochrane ROBINS-I (Risk of Bias in Non-randomized Studies of Interventions) tool [13], which is a risk of bias tool to assess the quality of non-randomized studies of interventions, was adopted to evaluate the methodological quality of each cohort-type study. The scoring system encompasses seven domains. The first two domains, covering confounding and selection of participants into the study, address issues before the start of the interventions that are to be compared (“baseline”). The third domain addresses the classification of the interventions themselves. The other four domains address issues after the start of interventions: biases due to deviations from intended interventions, missing data, measurement of outcomes, and selection of the reported result. The categories for risk of bias judgments are “low risk”, “moderate risk”, “serious risk”, “critical risk”, and “no information” when insufficient data are reported to permit a judgment.

In cases of randomized controlled trials (RCTs), the risk of bias was evaluated using the revised Cochrane Risk of Bias tool (RoB 2.0) [14]. According to this scoring system, seven domains were evaluated as “low risk of bias”, “high risk of bias”, or “unclear” according to the reporting on sequence generation, allocation concealment, blinding of participants, blinding of outcome assessment, incomplete outcome data, selective outcome reporting, and other potential threats to validity.

### 2.3. Statistical Analysis

The statistical analysis was performed using Review Manager (RevMan Version 5.4, Copenhagen, Denmark: The Nordic Cochrane Centre, The Cochrane Collaboration, 2020).

The primary outcomes of this study were the functional results after robotic fundoplication in patients suffering from GERD, hiatal hernia, or paraesophageal hernia compared to a laparoscopic approach. In addition, the differences among cases and controls were expressed as risk difference (RD) with pertinent 95% CI for dichotomous variables, to maintain analytic consistency and include all available data, according to Messori et al. [15]; the differences among cases and controls were expressed as mean difference (MD) with pertinent 95% confidence intervals (95% CI) for continuous variables. The risk difference represents the difference between the observed risks (proportions of individuals with the outcome of interest) in the two groups. If studies reported only the median, range, and size of the trial, the means and standard deviations were calculated according to Luo et al. and Wan et al. [16,17]. When studies reported only means for continuous variables and the sample size of the trial, a standard deviation was imputed, according to Furukawa et al. [18]. 

The overall effect was tested using Z scores, and significance was set at *p* < 0.05. Statistical heterogeneity between studies was tested by the Q statistic and quantified by the I^2^ statistic, a measurement of the inconsistency across study results and a description of the proportion of total variation in study estimates, that is due to heterogeneity rather than sampling error. In detail, an I^2^ value of 0% indicates no heterogeneity, 25% low, 25–50% moderate, and >50% high heterogeneity [19]. 

According to DerSimonian and Laird [20], the random-effects model was used for all analyses to account for the heterogeneity among included studies.

The presence of publication bias was investigated through a funnel plot, where the summary estimate of each study (Risk Difference) was plotted against a measure of study precision (Standard Error). In addition to visual inspection and the funnel plot, symmetry was tested using Egger’s linear regression method. [21] *p* values < 0.05 were considered statistically significant.

## 3. Results

### 3.1. Study Selection

A total of 339 articles were identified from electronic databases. After the removal of duplicate studies, 287 publications were screened according to the PRISMA flowchart (Figure 1).

Of the 72 articles that were selected for the title and abstract, 51 studies were excluded because they did not meet the inclusion criteria. Furthermore, the online full version of five articles was not available, and it was not possible to extract data from the abstract. The remaining 16 studies [22,23,24,25,26,27,28,29,30,31,32,33,34,35,36,37] were selected as they met the eligibility criteria and were included in the final analysis.

### 3.2. Baseline Characteristics of the Included Studies

This meta-analysis included 16 monocentric studies published between 2002 and 2021, involving 1064 patients suffering from GERD, hiatal hernia, or paraesophageal hernia, whereof 618 underwent laparoscopic and 445 robotic fundoplication, respectively. There were 4 RCT [27,32,33,34], 10 retrospective [22,23,24,25,26,28,29,30,35,36], and 2 prospective [31,37] trials. The number of patients ranged between 12 and 687, the mean age was between 3.8 and 72.5 years, and the mean BMI varied from 10.1 kg/m^2^ to 37.0 kg/m^2^. 

Major characteristics of the studies are shown in Table 1.

Intervention characteristics are described in detail in Table 2. Nissen fundoplication (360°) was performed in fourteen studies [22,24,25,26,27,28,30,31,32,33,34,35,36,37], Toupet fundoplication (270°) was reported in seven papers [23,25,29,31,35,36,37], Dor fundoplication (180°) was described in two articles [29,36], while in only one study [35] Watson partial anterior fundoplication was performed. Only two studies [22,36] included redo fundoplications.

### 3.3. Risk of Bias

The Cochrane RoB 2.0 and ROBINS-I tools were used to assess the quality of the included papers. 

Additionally, regarding the four randomized controlled trials, only one study [33] reported a low risk of bias. The other three studies [27,32,34] showed a high risk of bias, with the major bias due to deviations from the intended interventions, i.e., conversions to an open approach. Two conversions from laparoscopy were described by Draaisma et al. [27], and two conversions from the robotic approach were reported by Morino et al. and Nakadi et al. [32,34].

Due to the nature of the surgical interventions, blinding was impossible, but the results are unlikely to be affected by the lack of blinding. 

All non-randomized studies reported a risk of bias due to baseline confounding. Only three [22,26,31] authors with a consequent moderate risk of bias performed propensity score matching, while the other nine [23,24,25,28,29,30,35,36,37] had a severe risk of bias due to insufficient adjustment for confounding domains.

The results of the RoB 2.0 and ROBINS-I quality assessments are reported in Figure 2a,b and Figure 3a,b, which were created with Robvis (Risk-of-Bias VISualization), a web app that facilitates rapid production of publication-quality risk-of-bias assessment figures.

### 3.4. Primary Outcomes

The functional outcomes after laparoscopic (LF) and robotic fundoplication (RF) were analysed during a mean follow up period of 1-93.6 months, as described in Table 3. 

#### 3.4.1. 30-Days Readmission Rates

Only seven authors [22,23,25,26,33,35,37] reported 30-day readmission rates including 434 patients (207 RF and 227 LF) with no significant differences between the two groups [*p* = 0.73, RD = 0.00, 95% CI (−0.02, 0.03)]. No heterogeneity among the studies [Tau^2^ = 0.00; Chi^2^ = 0.85; df = 6 (*p* = 0.99); I^2^ = 0%] was reported (Figure 4).

#### 3.4.2. Persistence of Symptoms

Eleven studies [22,24,25,27,29,31,33,34,35,36,37] investigated the persistence of symptomatology after almost 1 month of follow-up. In addition, apart from two studies [25,37], which reported no ongoing symptoms, in the other nine [22,24,27,29,31,33,34,35,36], a total of 164 of the 641 patients referred reported lasting symptomatology without statistically significant differences between robotic and laparoscopic procedures (*p* = 0.60). Neither heterogeneity among the studies [Tau^2^ = 0.00; Chi^2^ = 6.19; df = 10 (*p* = 0.80); I^2^ = 0%] was described (Figure 5).

In particular, thirteen studies [22,23,24,26,27,30,31,32,33,34,35,36,37] reported the presence of postoperative dysphagia without significant differences between the two groups [39/387 RF and 47/529 LF, *p* = 0.77, RD = 0.00, 95% CI (−0.03, 0.02)]. Appendix A.

Seven authors [22,23,25,28,30,36,37] reported data regarding delayed gastric emptying: a total of 6 patients in each group had gastric paresis at follow-up, with no statistically significant differences between the two groups [6/215 RF vs. 6/242 LF; *p* = 0.99, RD = 0.00, 95% CI (−0.02, 0.02)]. Appendix A.

Only five articles [31,32,34,35,36] described the presence of postoperative pyrosis in 11 of 117 patients for the robotic group and 15 of 122 patients for the laparoscopic group, with no statistically significant differences between the two groups [*p* = 0.58, RD = −0.02, 95% CI (−0.09, 0.05)]. Appendix A.

#### 3.4.3. Recurrence 

In nine studies [22,24,25,27,30,31,33,34,37], recurrence of symptoms of reflux was described in 22 patients (5 RF and 17 LF), with no significant differences between the two groups [*p* = 0.36, RD = −0.02, 95% CI (−0.07, 0.03)]. The moderate heterogeneity among the studies [Tau^2^ = 0.00; Chi^2^ = 13.42; df = 8 (*p* = 0.10); I^2^ = 40%] was reported (Figure 6).

#### 3.4.4. Reoperation

A total of nine studies [22,25,27,30,31,33,34,36,37] reported reintervention during follow-up involving 22 patients with no significant differences between the two groups [*p* = 0.81, RD = 0.00, 95% CI (−0.04, 0.03)]. In addition, ten patients underwent reintervention after an initially successful robotic fundoplication: five experienced troublesome dysphagia [27,30,33,36], three had persistent GERD symptoms [36], one had an incisional hernia at the umbilicus [27], and another patient with a gastric torsion underwent a laparoscopic procedure with reduction of the torsion and fixation of the anterior gastric wall to the abdominal wall [34]. In the laparoscopic group, twelve patients were subject to reoperation during follow-up for persistent symptoms: six presented recurrent GERD symptoms [30,31,36] and six had severe dysphagia [27,30]. No heterogeneity among the studies [Tau^2^ = 0.00; Chi^2^ = 2.58; df = 8 (*p* = 0.96); I^2^ = 0%] was reported (Figure 7).

### 3.5. Publication Bias

It is recognized that publication bias can affect the results of meta-analyses; thus, we attempted to assess this potential bias using funnel plot analysis performed with Comprehensive Meta-analysis Software (CMA v.2). In evaluating all the analysed outcomes, we observed a symmetrical distribution of the studies without any publication bias by the Egger’s linear regression method (30-day readmission *p* = 0.84; recurrence *p =* 0.23; reoperation *p* = 0.60; persistence of symptomatology *p* = 0.12) (Appendix A).

## 4. Discussion

The functional disease of the esophago-gastric junction (EGJ) is a common health problem that often causes a serious deterioration of quality of life. Sometimes GERD patients do not achieve complete control of the symptoms with medical treatment with a proton pump inhibitor (PPI), needing surgical management. 

Up until now, laparoscopic fundoplication has been considered the gold standard for the surgical treatment of functional disease of the EGJ [3,4]. 

Additionally, after the first robotic-assisted Nissen fundoplication (RALF) [5], it has been debated if the robotic approach could improve surgical outcomes compared with the conventional laparoscopic fundoplication (CLF) [6], considering the documented safety and feasibility of the robot-assisted approach in this setting [7,8,9,10]. 

It could be rational to hypothesize advantages of robotic surgery to improve functional results; limitations of laparoscopic procedures due to lack of dexterity, lack of tactile sense, magnification of natural tremors, and two-dimensional visualization could be overcome.

However, the robotic technique presents an important limitation that is related to the high functional costs, as shown by Hartmann et al. [29], Morino et al. [32], and Albassam et al. [22], due to the instrumentation and reusable materials, the nursing costs, the investment costs, and the maintenance costs [34].

According to current literature, there is no clear evidence as to which minimally invasive surgical approach is superior for the treatment of functional diseases of the EGJ. In addition, to the best of our knowledge, this is the first meta-analysis reported on the comparative efficacy of available interventions in the management of functional diseases of the EGJ.

Several limitations must be considered in our study: first, because of the novelty of this topic, few studies are present in the literature. In fact, only 16 studies [22,23,24,25,26,27,28,29,30,31,32,33,34,35,36,37] published between 2002 and 2021 could be included in this meta-analysis, with a narrow sample size of 1064 patients suffering from GERD, hiatal hernia, or paraesophageal hernia.

Then, only four studies were RCTs [27,32,33,34] and two were prospective trials [31,37]. All the other ten included studies were retrospective [22,23,24,25,26,28,29,30,35,36]; the observed results in each study could be affected by many factors, such as standards in patients’ selection, the surgeon’s experience, or technical details. 

It is important to highlight that no one study had functional results as its primary outcome.

According to our results, both robotic and laparoscopic fundoplication are effective as well, reporting no significant differences between the two groups in terms of 30-day readmission rates (*p* = 0.73), lasting of symptomatology at almost 1 month of follow-up (*p* = 0.60), recurrence of symptoms of reflux (*p* = 0.36), and needing for reintervention during follow-up (*p* = 0.81). Moreover, two conversions from laparoscopy were described by Draaisma et al. [27], and two conversions from the robotic approach were reported by Morino et al. and Nakadi et al. [32,34]. Although nowadays the risk of conversion to open surgery has decreased due to higher surgeon expertise, it is important to underline that the conversion rate from robotic surgery is lower than that from laparoscopy, according to current literature [38,39,40]. 

Furthermore, regarding the persistence of symptomatology after 1 month from the intervention, no differences were found in postoperative dysphagia (*p* = 0.77), gastric paresis (*p* = 0.99) and postoperative pyrosis (*p* = 0.58). It is fair to specify that both persistence of symptomatology and recurrence could appear even after years with a treatment failure rate of 40%, as shown by Spechler SJ [41]. However, there is a lack of data concerning a follow-up longer than five years for both the medical and surgical approaches.

Even if, on the basis of our results, we can state that the robotic approach was effective and feasible for the surgical treatment of the functional disease of EGJ, we cannot declare any advantage on the basis of the functional analysis results. Both laparoscopic and robotic approaches could be selected to perform a Nissen fundoplication. On the other hand, the current literature presents a lack of ad hoc papers evaluating some important features, such as:-the functional outcomes obtained by a laparoscopic versus robotic approach.-data concerning a follow-up longer than 5 years for both medical and surgicalapproach.-indications and parameters for GERD-surgery.

The rationale is that robotic surgery should improve functional outcomes due to the magnified view and endowrist technology. However, it required making a call for future well-designed multicentre high-quality randomized controlled studies to evaluate the functional outcomes after robotic surgery for the treatment of functional disease of the EGJ, indications and parameters for GERD-surgery, and the long-term follow-up longer than 5 years. 

## Figures and Tables

**Figure 1 jpm-13-00231-f001:**
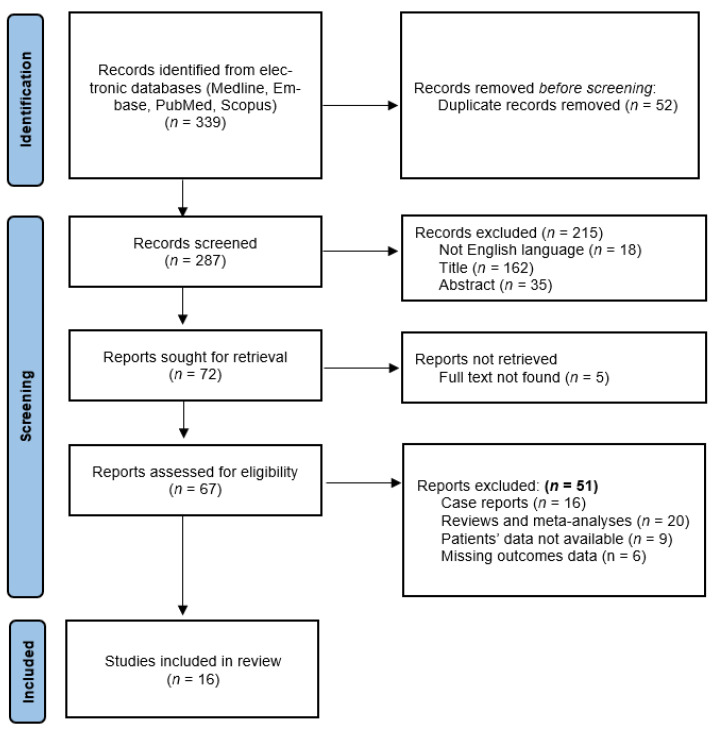
PRISMA 2020 flow diagram.

**Figure 2 jpm-13-00231-f002:**
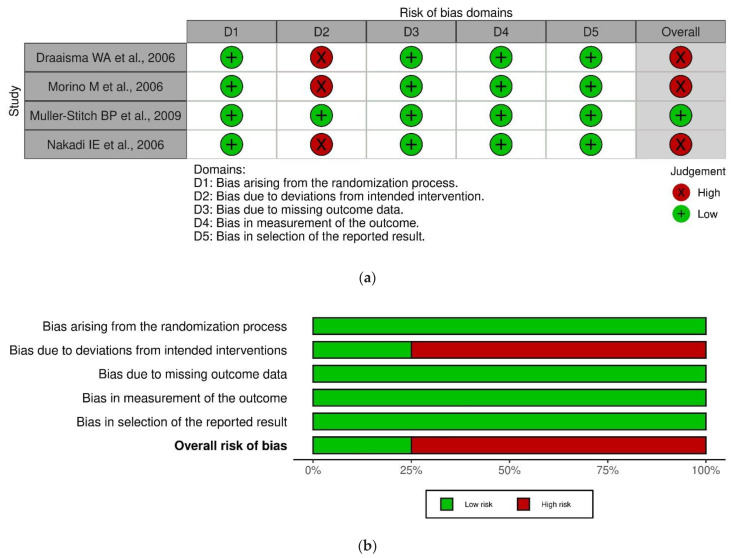
Summary of risk of bias for RCT studies [27,32,33,34].

**Figure 3 jpm-13-00231-f003:**
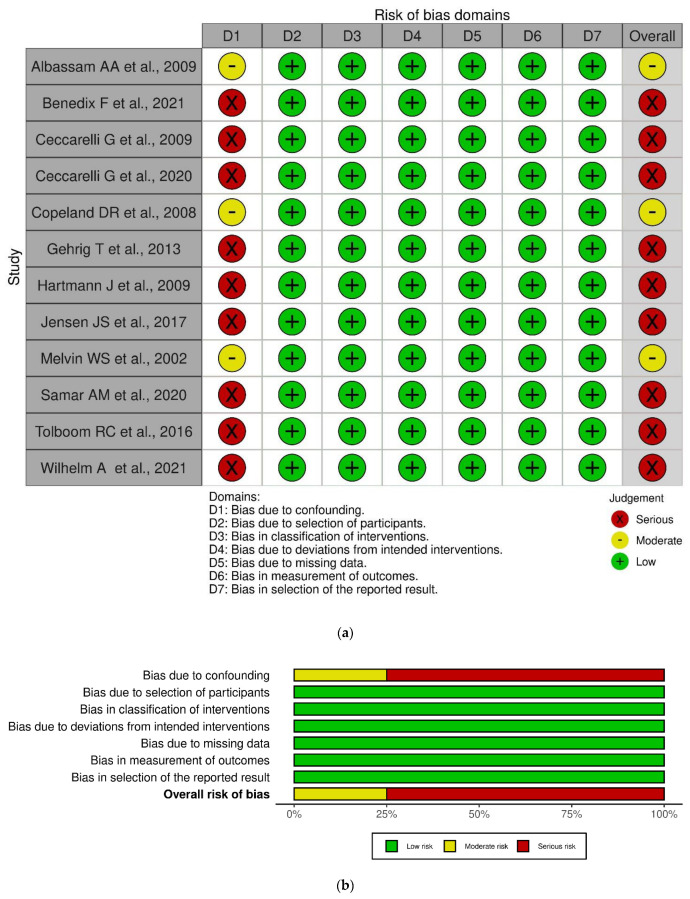
Summary of risk of bias for non-RCT studies [22,23,24,25,26,28,29,30,31,35,36,37].

**Figure 4 jpm-13-00231-f004:**
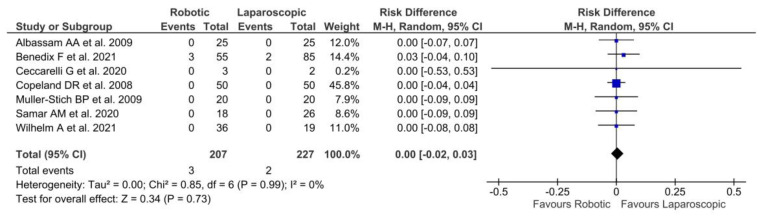
Forest plot of laparoscopic vs. robotic 30day readmission rates [22,23,25,26,33,35,37].

**Figure 5 jpm-13-00231-f005:**
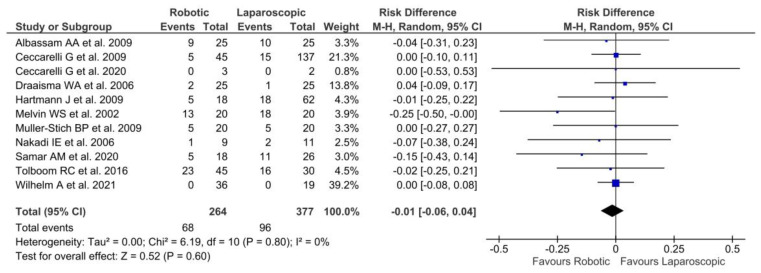
Forest plot of laparoscopic vs. robotic persistence of symptomatology [22,24,25,27,29,31,33,34,35,36,37].

**Figure 6 jpm-13-00231-f006:**
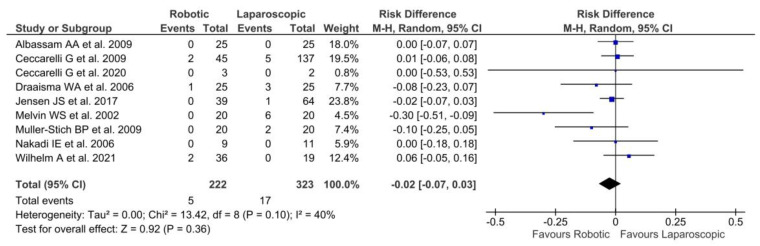
Forest plot of laparoscopic vs. robotic recurrence of reflux symptoms [22,24,25,27,30,31,33,34,37].

**Figure 7 jpm-13-00231-f007:**
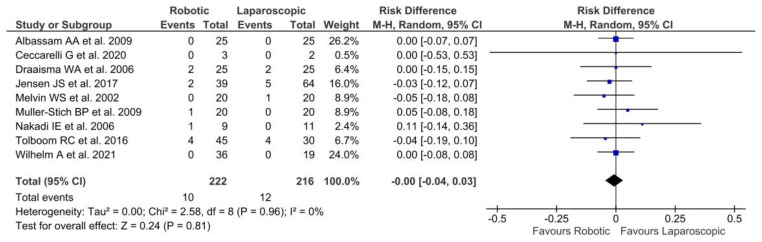
Forest plot of laparoscopic vs. robotic reoperation rates [22,25,27,30,33,34,36,37].

**Table 1 jpm-13-00231-t001:** Baseline characteristics of the included studies.

Study	Study Design	Study Period	Patients (n)			Male/Female (n)		Age (years) (SD)		BMI (kg/m^2^) (SD)		Diagnosis
			Total	Lap	Rob	Lap	Rob	Lap	Rob	Lap	Rob	
Albassam AA et al., 2009 [22]	Retrospective	Jan. 2005–Jul. 2008	50	25	25	9/16	11/14	3.8 (3.25)	5.4 (3.42)	10.9 (4.65)	10.1 (3.14)	GERD
Benedix F et al., 2021 [23]	Retrospective	Jan. 2016–Jul. 2020	140	85	55	29/56	18/37	62.9 (11.6)	63.5 (12.3)	29.3 (3.8)	29.5 (4.4)	Hiatal hernia and GERD
Ceccarelli G et al., 2009 [24]	Retrospective	Oct. 1992–Sep. 2007	183	137	45	-	-	52.5 (8.3)	55.0 (11.75)	27.0 (2.0)	28.0 (3.0)	GERD
Ceccarelli G et al., 2020 [25]	Retrospective	Dec. 2009–Dec. 2019	5	2	3	2/0	3/0	72.5 (13.5)	68.3 (14.2)	30.0 (2.6)	29.1 (5.6)	Giant Hiatal hernia
Copeland DR et al., 2008 [26]	Retrospective	1994–2005	100	50	50	-	-	8.9 (5.9)	9.75 (5.3)	33.0 (24.0)	37.0 (23.0)	-
Draaisma WA et al., 2006 [27]	RCT	Jan. 2003–Oct. 2005	50	25	25	17/8	16/9	50.5 (12.7)	39.0 (0.5)	31.0 (7.0)	27.0 (4.5)	GERD
Gehrig T et al., 2013 [28]	Retrospective	2003–2007	29	17	12	12/5	3/9	60.2 (11.8)	68.1 (7.9)	26.6 (4.4)	25.4 (2.6)	Paraesophageal hernia
Hartmann J et al., 2009 [29]	Retrospective	Jan. 2003–Dec. 2003	80	62	18	30/33	9/9	53.0 (14.0)	57.0 (13.0)	30.0 (4.7)	27.2 (4.3)	GERD
Jensen JS et al., 2017 [30]	Retrospective	Apr. 2013–Apr. 2015	103	64	39	23/41	18/21	49.4 (15.4)	52.0 (14.6)	26.9 (3.4)	26.5 (3.1)	GERD
Melvin WS et al., 2002 [31]	Prospective	-	40	20	20	7/13	13/7	49.6 (0.5)	42.9 (0.5)	-	-	GERD
Morino M et al., 2006 [32]	RCT	-	50	25	25	18/7	19/6	46.3 (11.3)	43.0 (12.8)	26.1 (2.3)	25.5 (2.9)	GERD
Muller-Stich BP et al., 2009 [33]	RCT	Aug. 2004–Dec. 2005	40	20	20	12/8	10/10	50.5 (12.4)	49.6 (12.0)	26.2 (3.4)	29.2 (5.83)	GERD
Nakadi IE et al., 2006 [34]	RCT	-	20	11	9	8/3	6/3	48.0 (4.0)	44.0 (4.0)	24.8 (0.7)	25.3 (1.2)	GERD
Samar AM et al., 2020 [35]	Retrospective	Jan. 2014–Jun. 2016	44	26	18	13/13	9/9	55.7 (8.2)	49.2 (7.75)	-	-	-
Tolboom RC et al., 2016 [36]	Retrospective	Jan. 2008–Dec. 2013	75	30	45	11/19	12/33	57.2 (2.2)	56.0 (2.2)	-	-	Hiatal hernia and GERD
Wilhelm A et al., 2021 [37]	Prospective	July 2015–June 2019	55	19	36	5/14	13/23	71.7 (13.5)	69.0 (11.5)	29.0 (6.5)	30.0 (3.5)	Hiatal hernia

**Table 2 jpm-13-00231-t002:** Intervention characteristics.

Study	De Novo Surgery		Redo Surgery		Nissen Fundoplication		Dor Fundoplication		Toupet Fundoplication		No Fundoplication		Operative Time (min)	
	Lap	Rob	Lap	Rob	Lap	Rob	Lap	Rob	Lap	Rob	Lap	Rob	Lap	Rob
Albassam AA et al., 2009 [22]	24	24	1	1	25	25	0	0	0	0	0	0	193.12 (26.6)	186.04 (21,1)
Benedix F et al., 2021 [23]	-	-	-	-	0	0	0	0	85	55	0	0	125.0 (35.5)	149.0 (42.1)
Ceccarelli G et al., 2009 [24]	-	-	-	-	137	45	0	0	0	0	0	0	86.2 (14.2)	65.0 (10.8)
Ceccarelli G et al., 2020 [25]	2	3	0	0	1	1	0	0	1	1	0	1	165.0 (5.0)	203.3 (17.8)
Copeland DR et al., 2008 [26]	50	50	0	0	50	50	0	0	0	0	0	0	107.0 (31.0)	160.0 (61.0)
Draaisma WA et al., 2006 [27]	25	25	0	-	25	25	0	0	0	0	0	0	115.0 (37.5)	125.0 (25.0)
Gehrig T et al., 2013 [28]	17	12	0	0	9	6	0	0	0	0	8	6	168.0 (42.0)	172.0 (31.0)
Hartmann J et al., 2009 [29]	62	18	0	0	0	0	62	18	0	0	0	0	116.0 (63.0)	207.0 (45.0)
Jensen JS et al., 2017 [30]	64	39	0	0	49	15	0	0	15	24	0	0	86.0 (19.0)	135.0 (27.0)
Melvin WS et al., 2002 [31]	20	20	0	0	17	17	0	0	3	3	0	0	101.7 (30.7)	160.2 (45.7)
Morino M et al., 2006 [32]	25	25	0	0	25	25	0	0	0	0	0	0	91.1 (10.6)	131.6 (18.3)
Muller-Stich BP et al., 2009 [33]	20	20	0	0	20	20	0	0	0	0	0	0	102.0 (19.0)	88.0 (18.0)
Nakadi IE et al., 2006 [34]	11	9	0	0	11	9	0	0	0	0	0	0	96.0 (5.0)	137.0 (12.0)
Samar AM et al., 2020 [35]	-	-	-	-	15	3	7 *	2 *	4	13	0	0	164.0 (43.0)	129.0 (22.0)
Tolboom RC et al., 2016 [36]	0	0	30	45	6	4	2	12	20	27	1	2	98.7 (6.2)	120.0 (2.5)
Wilhelm A et al., 2021 [37]	-	-	-	-	4	0	0	0	15	29	0	7	179.5 (42.0)	182.2 (6.9)

* Watson partial anterior fundoplication (120°).

**Table 3 jpm-13-00231-t003:** Functional outcomes.

Study	Mean Follow-Up (months)		30-days Readmission (n)		Persistence of Symptoms (n)		Delayed Gastrinc Emptying (n)		Pyrosis (n)		Dysphagia (n)		Recurrence (n)		Reoperation (n)	
	Lap	Rob	Lap	Rob	Lap	Rob	Lap	Rob	Lap	Rob	Lap	Rob	Lap	Rob	Lap	Rob
Albassam AA et al., 2009 [22]	19.25	19.25	0	0	10	9	1	2	-	-	1	0	0	0	0	0
Benedix F et al., 2021 [23]	3	3	2	3	-	-	0	0	-	-	8	5	-	-	1	1
Ceccarelli G et al., 2009 [24]	93.6	43.2	-	-	15	5	-	-	-	-	3	1	5	2	-	-
Ceccarelli G et al., 2020 [25]	6	6	0	0	0	0	0	0	-	-	-	-	0	0	0	0
Copeland DR et al., 2008 [26]	1	1	0	0	-	-	-	-	-	-	14	15	-	-	-	-
Draaisma WA et al., 2006 [27]	6	6	-	-	1	2	-	-	-	-	2	1	3	1	2	2
Gehrig T et al., 2013 [28]	-	-	-	-	-	-	2	0	-	-	-	-	-	-	0	1
Hartmann J et al., 2009 [29]	-	-	-	-	18	5	-	0	-	-	-	-	-	-	-	-
Jensen JS et al., 2017 [30]	22.5	26	-	-	-	-	0	0	-	-	4	2	1	0	5	2
Melvin WS et al., 2002 [31]	10.5	6.7	-	-	18	13	-	0	4	3	5	3	6	0	1	0
Morino M et al., 2006 [32]	12	12	-	-	-	-	-	-	0	0	0	0	-	-	-	-
Muller-Stich BP et al., 2009 [33]	12	12	0	0	5	5	-	-	-	-	0	1	2	0	0	1
Nakadi IE et al., 2006 [34]	12	12	-	-	2	1	-	-	0	0	0	0	0	0	0	1
Samar AM et al., 2020 [35]	1	1	-	-	11	5	-	-	2	2	4	2	-	-	-	-
Tolboom RC et al., 2016 [36]	11.7	3	-	-	16	23	3	3	9	6	3	6	-	-	4	4
Wilhelm A et al., 2021 [37]	28.7	31	0	0	0	0	0	1	-	-	1	1	0	2	0	0

## Data Availability

The data presented in this study are available on request from the corresponding author.

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
