# Peer review of "Robotic Surgery and Functional Esophageal Disorders: A Systematic Review and Meta-Analysis"

_jpm, 2023, doi:10.3390/jpm13020231_

Round 1

Reviewer 1 Report

Ms. No.: jpm 2032527

Title: Robotic surgery and functional esophageal disorders: a systematic review and meta-analysis

We commend the authors especially for the interesting topic of the manuscript. Definitely robotic  surgery for functional esophageal disorders represents a matter of debate that will continue in the upcoming years. The manuscript still has some limitations:

-        Please revise the introduction, the following paragraph: “Though, about 50% of GERD patients do not achieve complete control of the symptoms, needing for surgical management. [2]” ---this is an information that is completely updated – already in the early the number of laparoscopic fundoplications have dramatically decreased as we have noticed in the long term that a lot of patients have had recurrence of symptoms and the need of PPI despite surgery…….. Moreover your reference No 2 is from 1991!!!!! So please add an additional paragraph and use the past tense for the percentage of 50%

-        The authors state that: “conversions from laparoscopy were described by Draaisma et al. [27] and two conversions from robotic approach were reported by Morino et al. and Nakadi et al. [32,34]” – this coul not be interpreted nowadays when the risk of conversion to open surgery is exceptional. Please comment on this issue.?

-         In the last part of Fig 1 when you have 67 studies and you excluded 45 studies ---- you have included in the final analysis only 16 and NOT 21. Both the fig and the following text contain the same information and are very confusing !!!! Please clarify this part

-        Several times throughout the manuscript the authors discussed about recurrence of symptoms of reflux and concomitantly about lasting of symptomatology at almost 1 month of follow-up. This is a paradox as the recurrence could appear even after few years and to discuss this issue at “almost 1 month” is completely futile.

Minor modification:

-        The initial phrase from discussion is redundant with the initial phrase from the introduction.

-        In the discussion correct the word “adavanteges”

-        Please correct also “non one study”

-        Some phrases should be reformulated (eg. However The greatest limitation of robotic technique have to be remembered.)

In conclusion, we commend the authors for the manuscript, but additional information needs to be discussed. The manuscript in its current form needs MAJOR REVISION in order to be accepted.

Reviewer 2 Report

This is a nice systematic review and meta-analysis concerning GERD and its minimal invasive surgical procedures. The authors compare the classical laparoscopic fundoplication with robotic minimal invasive surgery. They could include 16 studies, of which inly two were randomized controlled studies. Overall, nearly all studies had significant risk of bias. Therefore the validity of the study is somehow questionable. One important point is that robotic surgery was not superior to classical laparoscopic surgery.

In total, the mean follow-up time was not very long. It is known from several studies, that recurrence after 4-5 years may be significant. The authors should comment on this issue. In total about 25% of patients complained of persisting symptomatology. This is also a well-known problem in selecting patients for fundoplication. But selecting parameters for surgery are not mentioned resp. studied. The authors should catch up for indications and parameters for GERD-surgery.

In total this study is surely eligible for publication in JPM after major revision.

Author Response

Pleese see the attachment.

Round 2

Reviewer 2 Report

Thank you for revising the manuscript.

The changes in the abstract are not understandable. The meaning of the sentence "About 50% of patients suffering of GERD need for surgical management several years ago.. Laparoscopic fundoplication has been considered the gold standard surgical treatment of functional disease of EGJ" remains completely unclear. What means EGJ? This abbreviation is not elucidated, which always should take place.

The same is applied to the corrected sentence in the introduction: what is meant by "several years ago" in this context?

In the discussion you state that medical treatment does not relieve the symptoms, so that surgery is needed. Is that the only reason for indicating surgery? This seems a bit undifferentiated: which hard criteria are  selected as indication for surgery?

It remains unclear which complications and issues are faced in the long run of patients suffering from GERD. Is there a solution in your eyes? There are many publications indicating that treatment (medical and/or surgical) still do show failure in the long run up to 40% of cases. You have to discuss these problems before advocating surgery, may it be conventional or robotic.

Author Response

"Please see the attachment". 
